# Dig into Detailed Structures: Key Context Encoding and Semantic-based Decoding for Point Cloud Completion

Somebody

## ABSTRACT

Recovering the complete shape of a 3D object from limited viewpoints plays an important role in 3D vision. Encouraged by the effectiveness of feature extraction using deep neural networks, recent point cloud completion methods prefer an encoding-decoding architecture for generating the global structure and local geometry from a set of input point proxies. In this paper, we introduce an innovative completion method aimed at uncovering structural details from input point clouds and maximizing their utility. Specifically, we improve both Encoding and Decoding for this task: (1) Key Context Fusion Encoding extracts and aggregates homologous key context by adaptively increasing the sampling bias towards salient structure and special contour points that are more representative of object structure information. (2) Semantic-based Decoding introduces a semantic EdgeConv module to prompt next Transformer decoder, which effectively learns and generates local geometry with semantic correlations from non-nearest neighbors. The experiments are evaluated on several 3D point cloud and 2.5D depth image datasets. Both qualitative and quantitative evaluations demonstrate that our method outperforms previous state-of-the-art methods.

## CCS CONCEPTS

• **Computing methodologies → Shape inference**.

## KEYWORDS

Point Cloud Completion; Generative Model; 3D-Context

**ACM Reference Format:**

Somebody. 2018. Dig into Detailed Structures: Key Context Encoding and Semantic-based Decoding for Point Cloud Completion. In *Proceedings of Make sure to enter the correct conference title from your rights confirmation emai (MM '24)*. ACM, New York, NY, USA, 10 pages. https://doi.org/XX

## 1 INTRODUCTION

With the popularity and development of 3D scanning equipment, 3D point clouds have wide applications in computer vision, such as autonomous driving [4], robotics [15], and industrial detection [19]. However, the original point clouds captured by 3D scanners or depth cameras in real-world are often incomplete due to self-occlusion, limited viewpoints, and low resolution. Therefore, recovering the complete and dense 3D shape from partial point clouds is important in many downstream 3D tasks [18, 20].

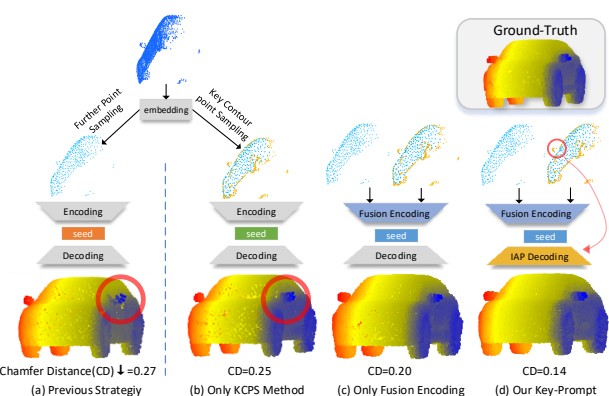

**Figure 1: Illustration of our main idea. We first compare the point cloud generated by a traditional down-sampling-based method (a) with that produced by our 'Key Contour Point Sampling' enhanced feature extraction (b). We further investigated the application of a fused encoding (c) and the semantic Interest Area Prompt (IAP) decoding (d), which also represents our proposed Key-Prompt model which provides more details and reduces the Chamfer distance metrics.**

Since deep neural networks have been successful in extracting features from 3D coordinates of point clouds, recent point cloud completion methods [14, 39] tend to use an encoding-decoding architecture for the coarse-to-fine generation. The encoding phase extracts features from the partial point cloud utilizing methods such as PointNet [22] or DGCNN [30]. A set of sparse points is generated via input features to sketch the global structure (called seeds), while the decoding phase generates a complete point cloud by refining the seed's local geometry. However, during the above processing, existing methods have two problems that still need to be solved: 1) The initially dense input point cloud needs to be aggregated several times during encoding, a process that often loses abundant geometric details. (2) Features are extracted from the incomplete input point cloud, posing challenges in directly generating the geometry of the correlated missing parts.

In the encoding phase, the necessary pre-processing downsampling strategy tends to abstract the spatial structure by merging a dense given point cloud into a component-level coarse point. Here, Farthest Point Sampling (FPS) [8] is a widely used method, which has an even sampling rate at each part, as illustrated in Fig. 1(a). To enhance the representativity of these point centers, methods vary the down-sample resolution or use different neighborhood scales for richer contexts. However, they neglected the fact that points that are located on contours or on complex structures often bring great help to the structural representation of the point cloud.

In the decoding phase, geometries of the missing parts are often recovered by constructing a relationship between sparse seed and input. The existing modeling of dependencies between seeds and

input points relies solely on neighboring geometric relationships, making it hard to learn long-distance correlation input information for seeds. Recent methods[17, 42] directly transformed the local features of the input part after self-attention coding into the features of missing part. AdaPoinTr[37] uses the geometry-aware module with both self-attention and cross-attention mechanisms to let seeds learn structural knowledge and detailed information. However, they pay more attention to the position and geometry embedding, and their usual correlations are built in the neighboring areas. In this case, those sparse seeds, especially of missing parts are hard to focus similarity local representation from incomplete input.

To solve the above problems, we propose a novel point cloud completion method with improved encoding-decoding architectures, as shown in Fig. 1. Firstly, a new adaptive down-sampling method is employed to maximize the preservation of key edge contours. Compared with FPS, the detailed geometries are well represented with the same ratio of down-sampled points (Fig. 1(b)). Then, two kinds of homologous down-sampling point cloud features are fused by a kind of self Vector Attention crosswise in our encoding phase to discover more input key context (Fig. 1(c)). After the encoding phase generates a set of seeds, we notice that semantic similarity is an essential cue for building local correlations between seeds and the partial point cloud, but it is always ignored. For example, similar semantic structures sometimes occur in symmetrical but distant locations. Thus, we design a semantic cross-attention decoding. This module initially acquires semantic structure for each seed proxy and then selectively acquires knowledge on the local geometry of missing parts via point-to-area attention learning as a prompt for the next transformer decoder. With richer feature extraction encoding and enhanced semantic guidance decoding, our method performs excellently for point cloud completion (Fig. 1(d)).

The main contributions of this work are summarized as follows:

- We propose a KCPS method that preserves richer geometry details via dynamically enforcing larger sampling biases toward key contour. Integrating KCPS and FPS extractor strategies, we transform given point clouds into input local proxies without sacrificing its contour descriptions.
- We propose a semantic-based IAP decoding method that introduces semantics cues into the dependency inference among given and inferred points, thus significantly improving point cloud completion with richer structure priors.
- Our method achieves state-of-the-art performance on various benchmarks, including 3D point cloud datasets such as PCN, Project shapenet55-34 and KITTI, as well as Redwood RGBD datasets.

## 2 RELATED WORK

### 2.1 Point Cloud Representation and Completion

Most of the early work in point cloud completion utilized 3D voxel grids [7, 13, 16, 25, 29] as an intermediate representation for each voxel block, which was then processed by methods such as 3D convolution. However, the voxelization will lose details of the point cloud, and the computational cost increases heavily in voxel resolution. The advent of PointNet revolutionized this landscape [22], enabling direct feature extraction from unstructured and disordered

point clouds using deep neural networks. Unlike PointNet, which operates on individual points, PointNet++ [23] leverages k-nearest neighbor search and maximum pooling operations to aggregate neighborhood context information, progressively abstracting local regions along hierarchical scales by repeatedly employing Farthest Point Sampling [8]. DGCNN [30] introduces the EdgeConv module, which maps the representation of edges into the feature space and dynamically aggregates point cloud neighborhoods multiple times. This dynamic graph structure enhances the network's ability to learn point cloud representations with semantics. PCN [39] is the first learning-based method for point cloud completion. After PCN, many methods [14, 27, 32, 42] adopt a similar coarse-to-fine architecture to enhance the real details of point cloud generation. SD-Net [3] introduces an end-to-end disentangled completion structure comprising two subnetworks for input refinement and missing part prediction, facilitating detail cross-recovery at various scales. Similarly, CRAPCN [24] incorporates a local attention mechanism during the encoding and decoding phase to fuse contextual information from multi-resolution point clouds. However, many of these methods incorporate various point contexts through different scales of down-sampling resolution or different numbers of neighbor aggregations, which often leads to overlooking detailed structures and crucial contour points. This oversight can have a significant impact on the representation and generation of whole structures.

### 2.2 Transformer in Point Cloud Completion

With the remarkable success of Transformer [28] in natural language processing, numerous applications have surged into the realm of computer vision. In the realm of point cloud completion, the unstructured set nature of point clouds prompted the introduction of Transformers to more effectively address the limitations of previous architectures in exploring associations within collections. Guo et al. [12] introduced PCT, which enhances the self-attention mechanism and incorporates local point cloud neighborhood information for embedding, making the Transformer more suitable for point cloud's vector attention. Therefore, many models based on the Transformer architecture accomplish detail completion by constructing relationships between seeds and inputs. Point Transformer combined the Vector Attention [41] with a U-Net style encoder-decoder framework. Seedformer [42] introduces a feature Up-Transformer module and the novel representation concept of patch seed, efficiently integrating spatial relations between neighboring points transformed into seeds. However, the self-attention decoding in Seedformer makes it hard to extract meaningful information from sparse seed points. Consequently, many Transformer-based models [2, 5, 17, 35] facilitate more detailed completion by establishing correlations between seeds and input data. PoinTr [37] and its variant [38] reformulate the point cloud completion problem as a set-to-set transformation and adopt a novel Transformer architecture to realize efficient completion. During the decoding phase, they utilize the geometry-aware module to establish a link between seed and input data, leveraging attention mechanism to learn implicit correlation from the input. However, due to the inherent disparities between input and seed representing missing parts, the set-to-set attention mechanism and current decoding architecture do not solve the problem of establishing associations between the two sets for interactive learning.

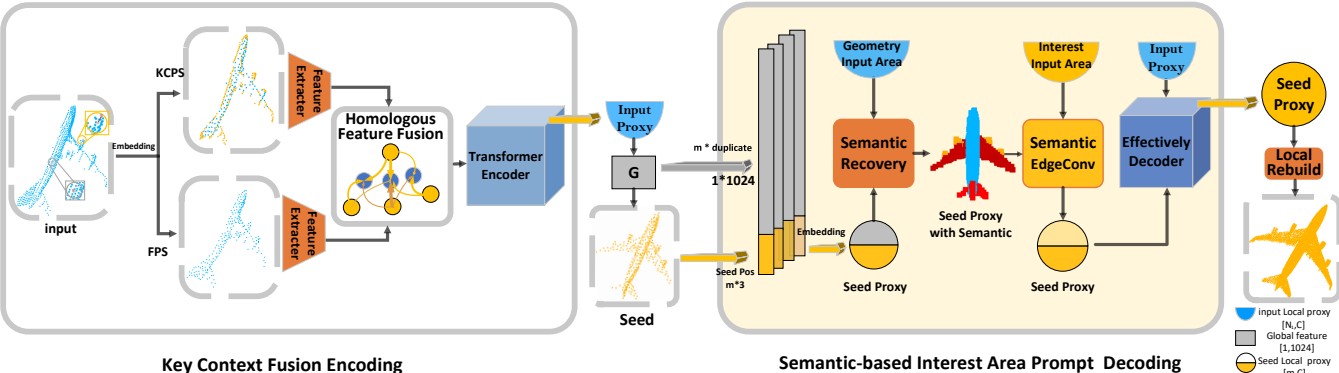

**Figure 2: The overall architecture of Key-Prompt. We use double extractors with FPS and KCPS to learn and fuse these homologous features with different focuses. While generating seeds, the architecture preserves semantic information of the feature extractors and the component association in the global structure as much as possible. The embedded semantic Edgeconv will prompt interest input areas for the seed proxy, facilitating richer eventual refinement.**

## 3 METHOD

### 3.1 Problem Formulation

There are multiple task representations of point cloud completion. Following the design of [37, 38], we define the point cloud complete task as a set-to-set transform task, which is well suited to the attention mechanism. Consider an input F-dimensional point cloud with N points, denoted by $\mathbf{P}_{in} = \{p_{in}^1, \ldots, p_{in}^N\} \subseteq \mathbb{R}^{N \times F}$, in the simplest setting of $F = 3$, each point contains 3D coordinates $p_{in}^i = (x_i, y_i, z_i)$. The final point cloud of the complete M points is represented as $\mathbf{Y} = \{y_1, \ldots, y_M\} \subseteq \mathbb{R}^{M \times F}$.

The overall network architecture of Key-Prompt is shown in Fig. 2, and we next introduce a few of the critical processing variables follow our architecture. To take advantage of the rich local details, Key Contour Point Sampling (KCPS) is used to construct a new point cloud $\mathbf{P}' \subseteq \mathbb{R}^{N_1 \times F}$ in parallel with the FPS point cloud $\mathbf{P} \subseteq \mathbb{R}^{N_1 \times F}$ for subsequent feature extraction and fusion. Here both the subset $\mathbf{P}'$ and $\mathbf{P}$ are down-sampled from input point cloud $\mathbf{P}_{in}$. Local points are converted into feature vectors representing a localized region, called input point proxy $\mathcal{F} = \{\mathcal{F}_1, \ldots, \mathcal{F}_{N_1}\} \subseteq \mathbb{R}^{N_1 \times C}$, which means that C-dimensional point cloud with $N_1$ down-sampled points. We transform the input point proxies into seeds $\mathbf{P}_q$ and seed proxies $Q$ via a global feature $\mathbf{G}$ for semantic preserving. The Interest Area Prompt decoding structure selects valid information from the interest input to enrich the representation of the seed proxy. Finally, we recover the local structure near each seed based on its proxy to generate the final point cloud $\mathbf{Y}$.

### 3.2 Key Context Fusion Encoding

*3.2.1 Key Contour Point Sampling.* Currently, down-sampling is often performed to aggregate fine-grained point clouds in the local vicinity during feature extraction. Traditional Farthest Point Sampling (FPS) [8] tends to result in the loss of points on delicate structures and contours, as shown in Fig. 3. Compared to flat surfaces, these points usually carry more structure information and local variations in computing neighbor edge vectors, which are particularly crucial for understanding an object's details and global structure.

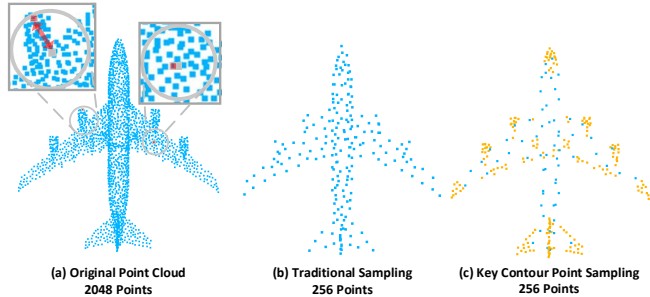

**(a) Original Point Cloud 2048 Points    (b) Traditional Sampling 256 Points    (c) Key Contour Point Sampling 256 Points**

**Figure 3: Our Sample Method. Figure (a) shows the difference between points on the contour and inside the structure calculating the offset; (b) and (c) are the results we obtained using traditional FPS method and our KCPS method.**

Such points are defined as 'Key Contour Points'. We have adapted and refined Ahmed's method [1], which defines contour points based on the offset from their neighborhood centers, allowing us to identify critical points more effectively. The neighborhood of a point $p_i$ is identified through a k-nearest neighbor search, represented as $V_i = \{n_1, n_2, \ldots, n_k\}$. Subsequently, the center coordinates $C_i$ of these neighboring points are calculated by incorporating each point $n_i$ within the vicinity:

$$C_i = \frac{1}{k} \sum_{j=1}^{k} n_j \tag{1}$$

To mitigate differences in density between regions and ensure scale invariance, the point dispersion distance $D_i$ of the region is calculated as follows:

$$D_i(V_i) = \max_{n \in V_i} \|p_i - n_i\| \tag{2}$$

Then, a degree of importance $I_i$ is defined to quantify the offset from the center while adaptive by vicinity densities as follows:

$$I_i = \frac{\|p_i - C_i\|}{D_i} \tag{3}$$

The input point cloud is sorted and sampled by the importance level $I_i$ to dynamically distinguish their differences in computing neighboring point edges. The top $\frac{q}{2}$ points are sampled as keypoints, for the remaining points, we use the FPS, stitching the two point clouds together to obtain the reconstructed point cloud $\mathbf{P}'$.

In contrast to Ahmed's method [1], we use the maximum value in calculating $D_i$ to achieve greater robustness, especially with non-uniform point distributions. We also find that calculating too few neighborhood points may only select points to outline shapes, while too many can obscure critical structural areas. A moderate number of neighbors is often best for identifying salient contour keypoints. Fig. 3 illustrates that our method results in more accurately capturing intricate structures with the same sampling ratio.

*3.2.2 Double Extractor and Context Fusion.* Aimed at capturing both the broad structure and intricate details, we propose a dual-extractor framework depicted in Fig. 2. Starting with the embedded input point cloud, two sampling techniques are applied for the next feature extraction. We utilize FPS to obtain coordinates $\mathbf{P}$, which helps in understanding the overall shape, while KCPS acquires coordinates $\mathbf{P}'$, focusing on structure details. Then, points and their corresponding embedded features are fused to ensure that the detail-focused insights are integrated with the overall view provided.

In each feature extractor, we follow DGCNN [30], first build a local graph structure based on local neighborhoods for the point cloud, noting the edge vector connecting the neighbor $p_j$ of the $p_i$ as $p_j - p_i$. We connect the edge vector with $p_i$ and feed them into the neural network. The steps are as follows:

$$\mathcal{F}_{ij} = \text{ReLU}\left(Liner\left(\mathbf{x}_i, \mathbf{x}_j - \mathbf{x}_i\right)\right) \tag{4}$$

$$\mathcal{F}_i = Max(F_{p_{ij}}), \forall j : p_j \in \kappa\left(p_i\right) \tag{5}$$

where $\mathbf{x}_i, \mathbf{x}_j$ represents the point's embedded feature, we gather the features using max-pooling as $Max$ for whose coordinates $p_j$ locates within the k-neighborhood of $p_i$, represented as $\kappa(p_i)$. For ease of writing, we combine these two EdgeConv steps and write them in a formula as follows:

$$\mathcal{F}_i = \text{EdgeConv}(x_i, x_j), \forall j : p_j \in \kappa\left(p_i\right) \tag{6}$$

It is worth emphasizing here that similar geometrical structures will yield similar semantics. EdgeConv is actually a dynamic graph attention mechanism based on subtraction relations. Our dual-extraction strategy, using multiple EdgeConv layers, conducts a secondary sampling based on each distinct sampling method, thereby dynamically updating the graph structure independently. This process not only refreshes the information in the salient structure but also dynamically disseminates details and semantic information throughout the point cloud.

We innovatively use Vector Attention [41] to fuse KCPS and FPS-derived features and coordinates. Vector Attention was initially used for self-attention in one point cloud, but our two point clouds are homologous which makes it easy to extend it to cross-point cloud feature fusion. Instead of learning each point in the set globally, we refine each point proxy $\mathcal{F}_i$, based on its local neighborhood spatial coordinates $p_j'$ and corresponding proxy $\mathcal{F}_j$, enhancing the structural comprehension of the whole point cloud. Compared to the dot product, the $\beta$-subtracted relational function are better reflection. Relative position coding denoted as $\delta$ is additionally added

into fusion:

$$\delta = \text{Liner}(p_i - p_j'), \forall j : p_j' \in \kappa\left(p_i\right) \tag{7}$$

$$\mathcal{F}_i = \rho\left(\gamma\left(\beta\left(\phi\left(\mathcal{F}_i\right), \psi\left(\mathcal{F}_j\right)\right) + \delta\right) \odot \alpha\left(\mathcal{F}_j + \delta\right)\right) \tag{8}$$

where $\phi, \psi$, and $\alpha$ are linear projection functions, and the mapping function $\gamma$ is an MLP with two linear layers and one ReLU nonlinearity. $\rho$ is a normalization function such as softmax. Therefore, we apply the neighbor area attention mechanism to fuse each query point cloud $p_i$ with $p_j'$, while incorporating more locally informative features $F_j$ into final input local proxy $\mathcal{F}_i$.

### 3.3 Semantic-based Decoding

In this section, we innovatively introduce an Interest Area Prompt Decoding shown in Fig. 2 that significantly helps seed to achieve selective learning of relevant information in the input. We define the seed proxy as $Q$, incomplete input feature as $\mathcal{K}, \mathcal{V}$, and formulate the Transformer encoder $\mathcal{M}_E$ and decoder $\mathcal{M}_D$ layers processes as follows:

$$\mathcal{V} = \mathcal{M}_E(\mathcal{F}), \quad \mathcal{H} = \mathcal{M}_D(Q, \mathcal{V}) \tag{9}$$

The current decoding method [5, 38] only builds correlations based on geometric distances, ignoring the missing parts' seeds to focus on the most relevant areas, as shown in Fig. 4 (b). Our method recovers the seed semantic information shown in Fig. 4 (c) while successfully focusing on the input empennage and wing located on the symmetric side using the similarity of the semantic features

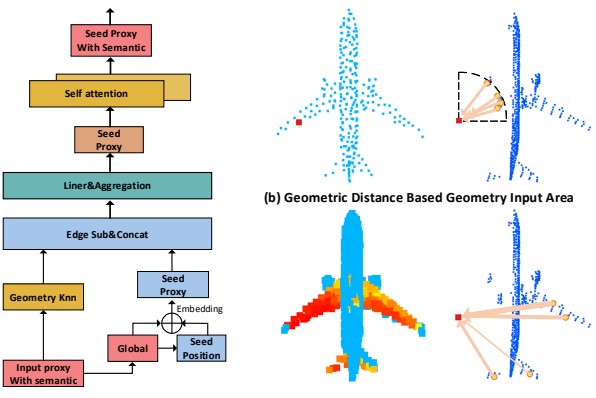

**(b) Geometric Distance Based Geometry Input Area**

**(a) Semantic Recovery Module Details**

**(c) Semantic Structure Based Interest Input Area**

**Figure 4: We show details of semantic recovery module and the difference of input Area of interest for geometric-based and semantic-based EdgeConv.**

We detail the process of recovering potential semantic information, utilizing geometric and self-attention mechanisms, in Fig. 4 (a). Each input local proxy are initially through Transformer Encoder and embedding, subsequently aggregating them into global features denoted as $\mathbf{G}$. Following this step, we proceed to generate a sparse seed $\mathbf{P}_q$ along with its corresponding seed proxy, represented as $Q$:

$$Q_i = [\mathbf{G}, p_q^i] = [Max(\mathcal{V}), p_q^i] \tag{10}$$

The input proxy $\mathcal{F}$ and global features $\mathbf{G}$ could encompass semantic structures, thanks to our dynamic EdgeConv extractor's capability to infuse features with semantic significance. This principle is rooted in observation that regions with geometrical similarities

also tend to exhibit similar local neighborhood graph structures during feature extraction. Semantic similarity is evident in geometric structures found within one component or between similar components, such as empennage and the wings of an airplane. Then, using geometric distance-based EdgeConv, we quickly capture the semantic features of neighboring inputs, aiding in the retrieval of semantic information from the input:

$$\phi(Q_i) = \text{EdgeConv}(Q_i, \mathcal{K}_j - Q_i), \forall j : p_k^j \in \kappa\left(p_q^i\right), \quad (11)$$

Further, we use self-attention to assist in self-similar structural associations originating from global features $\mathbf{G}$, with the ability to capture long-range dependencies:

$$Q = \mathcal{M}_E(Q) \quad (12)$$

In this way, points in the same or related structure can learn similar semantics from each other. We use the distance in the feature space to measure the degree of semantic similarity, as shown Fig. 4 (c). So we could use seed local proxy $Q$ with semantic information to construct new graph structure relations with an interest area of input proxy $\mathcal{K}$ in the feature space:

$$\tau(Q_i) = \text{EdgeConv}(Q_i, \mathcal{K}_l - Q_i), \forall l : K_l \in \kappa(Q_i), \quad (13)$$

This brings the semantically corresponding partial features closer to missing parts, which amounts to a directed cue to the decoder of the subsequent set to set. So we call it Interest Area Prompt (IAP). We then perform a more focal and effective Transformer decoder of $Q$ and $\mathcal{V}$, as shown in Eq. 9.

Afterwards, we refine the sparse seed into a dense and complete point cloud by converting the seed proxy $Q$ into the displacement deviation $\mathcal{H}$ of the neighborhood point through a simple MLP module. The final point cloud is represented as:

$$Y_i^k = \mathcal{H}_i^k + p_q^i, k = \frac{M}{m} \quad (14)$$

where $m$ and k are the number of seed points and each seed localized points, resulting in a point cloud $\mathbf{Y} \subseteq \mathbb{R}^{M \times 3}$ of M points.

## 3.4 Loss Function

Fan et al. [9] proposed the Chamfer Distance (CD) as an evaluation metric for the 3D reconstruction task:

$$d_{\text{CD}}(P_1, P_2) = \frac{1}{P_1} \sum \min_{y \in P_2} \|x - y\|_2^2 + \frac{1}{P_2} \sum \min_{x \in P_1} \|y - x\|_2^2 \quad (15)$$

The CD distance is used to measure the nearest-square distance between the predicted point cloud $P_1$ and the ground-truth point cloud $P_2$, which is computed for each of the two point cloud sets and averaged.

To ensure the sparse seed $\mathbf{p}_q$ adequately represents the ground truth shape, the ground truth is down-sampled to match the seed point count denoted by $Y_{gt}^1$. This step allows for the calculation of the loss $\mathcal{L}_s$ by comparing $\mathbf{p}_q$ with $Y_{gt}^1$. For assessing the quality of the finally refined generated result, the comparison with the ground truth generates the final result loss, $\mathcal{L}_f$. In addition, following the denoising strategy from AdaPoinTr [38], noise is introduced to the seed points during training. The goal is to predict the distribution of these noise to ground truth values in order to make the Transformer decoder more robust to the initial seed. The CD between the noise-adjusted generation and the corresponding amount of ground truth,

denoted by $Y_{gt}^2$, is then evaluated to quantify the noise elimination effectiveness, resulting in the loss $\mathcal{L}_d$:

$$\mathcal{L}_s = d_{\text{CD}}(p_q, Y_{gt}^1), \mathcal{L}_f = d_{\text{CD}}(p_q, Y_{gt}), \mathcal{L}_d = d_{\text{CD}}(p_q, Y_{gt}^2) \quad (16)$$

The training loss function $\mathcal{L}$ is finally defined as:

$$\mathcal{L} = \mathcal{L}_s + \mathcal{L}_f + \lambda \mathcal{L}_d \quad (17)$$

## 4 EXPERIMENTS

In this section, we conducted experiments on point cloud completion, including real-world scenes, with validation on more challenging 2.5D depth images. We compared our approach to baseline methods on benchmark datasets like PCN and the more challenging Project-ShapeNet-55/34. We also pre-trained our model on PCN and tested it on real-world datasets like KITTI and Redwood RGBD, outperforming existing approaches both qualitatively and quantitatively.

## 4.1 Experimental Settings

*4.1.1 Implementation Details.* We implement Key-Prompt with Pytorch [21]. All models are trained using two NVIDIA Tesla V100 gpus. The ablation study and complexity analysis are conducted under the same experimental conditions. For the training phase, we utilize AdamW for optimizing our model, setting the initial learning rate to $10^{-4}$, and 400 epochs will be conducted with a learning rate decay set to 0.9. Moreover, considering that the local neighborhood of the KCPS point cloud is denser and richer, we incorporate a larger number of neighbors (24) to extract meaningful key context for feature extraction while using 16 neighbors for point clouds of FPS.

*4.1.2 Evaluation Metrics.* We adhere to prior research practices and utilize the average Chamfer distance(CD) as our primary evaluation metric, which is widely referenced in the field. As mentioned in section 3.4, a higher average CD indicates a greater disparity between the two point cloud sets. Specifically, the CD significantly increases when the generated point cloud contains extra or missing parts compared to the ground truth. Additionally, we employ the F-score metric, following the methodology introduced by Tatarchenko et al. [26], to evaluate the proportion of similar components. This metric gauges similarity by assessing whether a point from one set falls within a certain threshold of another set.

## 4.2 Point Cloud Completion

*4.2.1 Evaluation on PCN Dataset.* **Data.** The PCN dataset [39] is one of the most commonly used datasets in the field of point cloud completion. It is a subset of ShapeNet including 8 categories with a total of 30974 CAD models. To emulate real-world sensor data, incomplete point clouds were generated by back-projecting 2.5D depth images from 8 viewpoints. These created incomplete point clouds consist of 2048 points, while the ground truth contains 16,384 points sampled from the complete point cloud. We adopt the identical experimental setting as PCN [39] to ensure a fair comparison with existing models.

**Results.** We provide Chamfer Distance and F-Score@1% results for the L1 norm (×1000) in Tab. 1, comparing recent models. Our method consistently outperforms prior state-of-the-art methods

**Table 1: Completion results on PCN dataset in terms of per-point L1 Chamfer Distance ×1000 (lower is better) and F1-score.**

|  | Plane | Cabinet | Car | Chair | Lamp | Couch | Table | Boat | CD-$\ell_1$ | F-Score@1% |
|---|---|---|---|---|---|---|---|---|---|---|
| FoldingNet [36] | 9.49 | 15.80 | 12.61 | 15.55 | 16.41 | 15.97 | 13.65 | 14.99 | 14.31 | 0.322 |
| TopNet [27] | 7.61 | 13.31 | 10.90 | 13.82 | 14.44 | 14.78 | 11.22 | 11.12 | 12.15 | 0.503 |
| AtlasNet [11] | 6.37 | 11.94 | 10.10 | 12.06 | 12.37 | 12.99 | 10.33 | 10.61 | 10.85 | 0.616 |
| PCN [39] | 5.50 | 22.70 | 10.63 | 8.70 | 11.00 | 11.34 | 11.68 | 8.59 | 9.64 | 0.695 |
| GRNet [34] | 6.45 | 10.37 | 9.45 | 9.41 | 7.96 | 10.51 | 8.44 | 8.04 | 8.83 | 0.708 |
| PoinTr [37] | 4.75 | 10.47 | 8.68 | 9.39 | 7.75 | 10.93 | 7.78 | 7.29 | 8.38 | 0.745 |
| NSFA [40] | 4.76 | 10.18 | 8.63 | 8.53 | 7.03 | 10.53 | 7.35 | 7.48 | 8.06 | - |
| PMP-Net++ [31] | 4.39 | 9.96 | 8.53 | 8.09 | 6.06 | 9.82 | 7.17 | 6.52 | 7.56 | - |
| SnowflakeNet [33] | 4.29 | 9.16 | 8.08 | 7.89 | 6.07 | 9.23 | 6.55 | 6.40 | 7.21 | - |
| SeedFormer [42] | 3.85 | 9.05 | 8.06 | 7.06 | 5.21 | 8.85 | 6.05 | 5.85 | 6.74 | - |
| ProxyFormer [17] | 4.01 | 9.01 | 7.88 | 7.11 | 5.35 | 8.77 | 6.03 | 5.98 | 6.77 | - |
| AdaPoinTr [38] | 3.68 | 8.82 | 7.47 | 6.85 | 5.47 | 8.35 | 5.80 | 5.76 | 6.53 | 0.845 |
| **Key-Prompt** | **3.43** | **8.70** | **7.30** | **6.56** | **5.05** | **8.20** | **5.76** | **5.55** | **6.32** | **0.857** |

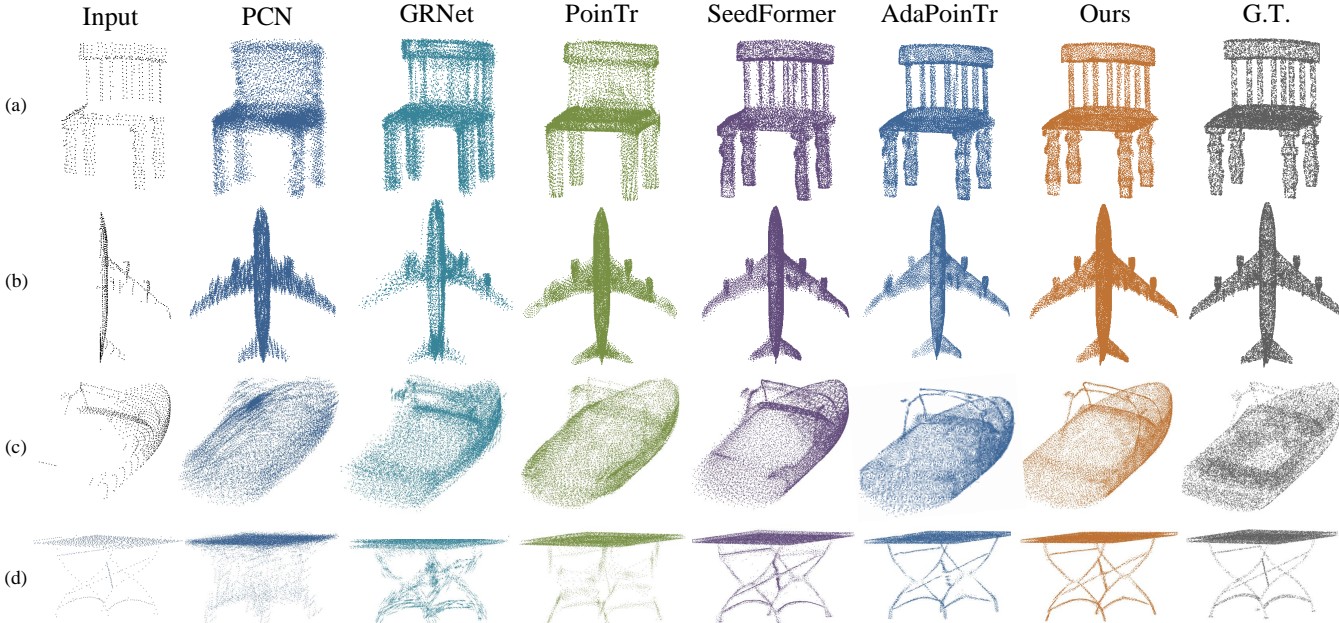

**Figure 5: Qualitative comparisons on the PCN dataset are shown. Each method completes the object from the incomplete point cloud shown at the far left of each row. Our method produces completion that most closely matches the ground truth.**

across all categories, with notably superior average metrics. Additionally, in Fig. 5, we visually compare shapes from chair, plane, boat, and table categories with SeedFormer [42], AdaPoinTr [38], and other methods[34, 37, 39]. These comparisons highlight Key-Prompt's ability to generate highly specific and realistic results with richer detailed information, especially for missing parts. For instance, as seen in the chair backrest example in Fig. 5(a), our method delineates a much clearer structure. *More visualization results and shows of semantic interest areas are available in the Appendix.*

*4.2.2 Evaluation on Project ShapeNet-55/34.* **Data.** The Project-ShapeNet-55 dataset encompasses a wider array of object categories and introduces different methods of simulating incomplete point clouds. It incorporates the entirety of ShapeNet's 55 categories,

where both training and testing are conducted on this comprehensive set. The unique aspect of Project-ShapeNet-55 is its method of generating partial inputs, specifically through a noisy back projection technique. This process involves rendering each sample from 16 randomly selected viewpoints to create depth images. Subsequently, these depth images are utilized to simulate incomplete point clouds by employing a noisy back projection method, effectively replicating scenarios encountered in the real world. The Project-ShapeNet-34 dataset is designed to test the generalization capabilities of various models. Utilizing 34 categories for training purposes sets up a unique challenge for models to accurately predict an additional unseen 21 categories during the training phase while the remaining design is consistent with Project-ShapeNet-55.

**Table 2: Completion results on Project-ShapeNet-55 dataset**

| | Many-Shot categories | | | | | Few-Shot categories | | | | | | |
|---|---|---|---|---|---|---|---|---|---|---|---|---|
| | Table | Chair | Airplane | Car | Sofa | Microwaves | Bag | Pillow | Keyboard | Rocket | CD-$\ell_1$ | F-Score |
| PCN [39] | 14.79 | 15.33 | 9.07 | 12.85 | 17.12 | 39.54 | 18.64 | 20.04 | 13.69 | 10.98 | 16.64 | 0.403 |
| TopNet [27] | 14.40 | 16.29 | 9.85 | 13.61 | 16.93 | 26.64 | 18.69 | 19.57 | 11.05 | 10.45 | 16.35 | 0.337 |
| GRNet [34] | 12.01 | 12.57 | 8.30 | 12.13 | 14.36 | 18.35 | 14.67 | 15.15 | 9.71 | 8.58 | 12.81 | 0.491 |
| SnowflakeNet [33] | 10.49 | 11.07 | 6.35 | 11.20 | 12.59 | 16.92 | 12.86 | 14.02 | 8.12 | 7.49 | 11.34 | 0.594 |
| PoinTr [37] | 9.97 | 10.43 | 6.02 | 10.58 | 12.11 | 17.06 | 12.15 | 12.57 | 7.61 | 6.86 | 10.68 | 0.615 |
| AdaPoinTr [38] | 8.81 | 9.12 | 5.18 | 9.77 | 10.89 | 17.43 | 10.93 | 11.82 | 6.79 | 5.58 | 9.58 | 0.701 |
| Key-Prompt | **8.69** | **8.96** | **5.13** | **9.60** | **10.77** | **13.64** | **10.06** | **11.21** | **6.64** | **5.44** | **9.35** | **0.714** |

**Table 3: Completion results on Project-ShapeNet-34 dataset**

| | 21 unseen categories | | | | |
|---|---|---|---|---|---|
| | Pillow | Skateboard | Earphone | CD-$\ell_1$ | F-Score |
| PCN [39] | 23.45 | 17.27 | 24.82 | 21.44 | 0.307 |
| TopNet [27] | 17.55 | 12.59 | 19.34 | 15.98 | 0.358 |
| GRNet [34] | 18.64 | 10.60 | 15.00 | 15.03 | 0.439 |
| SnowflakeNet [33] | 15.35 | 9.58 | 15.19 | 12.82 | 0.551 |
| PoinTr [37] | 14.82 | 8.98 | 14.23 | 12.43 | 0.555 |
| AdaPoinTr [38] | 13.72 | 8.34 | 12.30 | 11.37 | 0.642 |
| Key-Prompt | **13.32** | **7.43** | **11.39** | **10.97** | **0.654** |

For the evaluation strategy, we align with the methodology described by Yu et al. [37], where the ground truth point clouds are represented by 8,192 points. The generated incomplete point clouds used as input are configured to contain 2,048 points.

**Results on Projected-ShapeNet-55** The introduction of noise into the input point cloud contributes to the dataset's complexity and realism, thus presenting a heightened challenge for completion. We provide an analysis of our models and existing methodologies, outlining the class-specific Chamfer Distance (CD) and overall CD for each approach. Specifically focusing on ten categories, we present detailed results. Tab. 2 demonstrates that Key-Prompt outperforms other approaches across these categories and in terms of overall CD. The Many-Shot categories on the left have more than 2500 train samples, while the Few-Shot categories contain less than 80 train samples. Our results greatly outperform AdaPoinTr [38], suggesting that our method is able to mine more valid information from fewer training sets.

**Results on Projected-ShapeNet-34** In Tab. 3, we present the performance of our model and compare it with existing models. The data clearly indicate that our model outperforms others in all unseen categories. Importantly, our model not only could improve results in familiar categories but also significantly better identify unseen categories, such as Pillow, Skateboard, and Earphone. This performance highlights our model's strong generalization ability, showing its effectiveness in classifying new object types that were not included in the training dataset.

## 4.3 Real Scene Point Cloud Completion

*4.3.1 Evaluation on KITTI.* **Data.** To evaluate our model's performance with real-world data, we conducted experiments on the KITTI [10] dataset sourced from LIDAR scans. The KITTI dataset, widely recognized in autonomous driving research, presents a challenge due to the sparsity of LIDAR-derived data. Thus, generating complete and dense point clouds is crucial for downstream tasks, such as 3D target detection. Specifically, we extracted and localized the point clouds of car objects as input by employing a 3D bounding box within each frame. Since this dataset does not have a complete point cloud as ground truth, we follow the GRNet [34] using Fidelity Distance (FD) and Minimal Matching Distance (MMD) as evaluation metrics.

**Results.** Following the GRNet's [34] setup, we pre-trained our model on the Car category of the PCN [39] dataset to complement the incomplete cars in KITTI. As shown in Tab. 4, our model outperforms several baseline models. The car point clouds generated by our model accurately depict windows and tires while capturing intricate details, such as reflectors, as evidenced in Fig. 6.

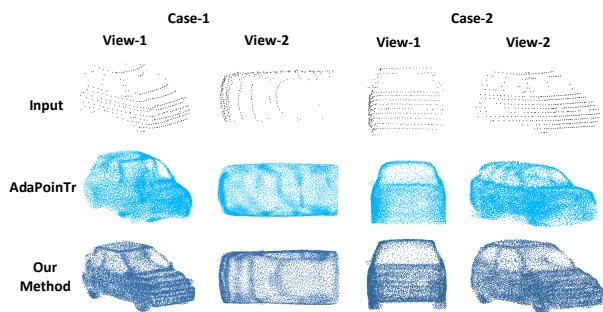

**Figure 6: Qualitative results on the KITTI dataset. We show two different views of each object while our method can recover a car with more accurate contour and details.**

*4.3.2 Depth Image Completion to 3D.* Exploring the efficacy of models pre-trained on the PCN [39] dataset, we engaged with real object scans from the Redwood 3D Scans [6] dataset, focusing on localized shapes of chairs, sofas, and cars derived from depth images captured with an RGB-D camera. Unlike LiDAR data, point clouds from real-depth images typically suffer from lower accuracy, more noise, and increased occlusion complexities. Given the scarcity of complete meshes in the Redwood 3D Scans dataset, our evaluation was qualitative, assessing the model's ability to recover original object shapes and details with high fidelity. The experimental results in Fig. 7, illustrate that our predictions demonstrate superior recovery of the original object shape and local details.

**Table 4: KITTI Dataset results. The comparison between the following models is based on the FD and MMD metrics.**

| CDl2(x1000) | AtlasNet [11] | PCN [39] | FoldingNet [36] | TopNet [27] | PFNet [14] | GRNet [34] | SeedFormer [42] | AdaPoinTr [38] | **Key-Prompt** |
|---|---|---|---|---|---|---|---|---|---|
| Fidelity | 1.759 | 2.235 | 7.467 | 5.354 | 1.137 | 0.816 | 0.151 | 0.237 | **0.136** |
| MMD | 2.108 | 1.366 | 0.537 | 0.636 | 0.792 | 0.872 | 0.516 | 0.392 | **0.380** |

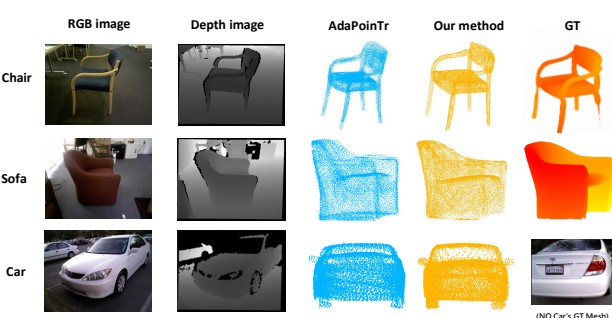

**Figure 7: Results of the chair, sofa, and car in Redwood. We show the input point cloud and ground truth, as well as the prediction results from AdaPoinTr and our method.**

## 4.4 Ablation Study

In this section, we demonstrate the effectiveness of several module designs in Key-prompt. We replace our method with some traditional modules for ablation experiments, and all our networks are trained and tested on PCN datasets with the same settings.

**Sampling Methods.** We take different sampling methods for feature extraction. We perform separate (A) Farthest Point Sampling and (A1) Key Contour Point Sampling, and we find that using KCPS alone gives only a slight boost but could better recover some detailed structures. Therefore, we utilize both sampling methods as follows: (B) Certain key contour points are directly incorporated into the output of the farthest point sampling as spatial fusion, and they persist as input for the subsequent Feature Extractor.

**Fusion Encoding.** Our other strategy is to use two feature extractors to take the two sampling methods separately and fuse the output features. We have alternative fusion schemes for feature fusion:(C) Feature fusion using traditional attentional mechanism fusion and (D) Vector Attention (VA) based feature fusion method. Experiments have shown that Vector Attention mechanism gives better results. We found that the method of VA fusion originally used for one point cloud can be better applied to our type of fusion of homologous point clouds and features.

**Interest Area Prompted Decoding.** Traditional point cloud completion models based on transformer cross-decoder use the basic set to set Transformer in the decoding phase. We designed the IAP module to change this decoding structure, which allows for models with fewer parameters. Results in Tab. 5 show clear improvement of our IAP design (6.32 vs 6.53).

## 4.5 Accuracy-Complexity Trade-Offs

Our dual-feature extractor architecture, operating in parallel, adds little to the computational complexity. Thanks to the IAP module, which significantly enhances the efficiency of the Transformer decoder architecture by sharpening the focus of attention. We are

**Table 5: Ablation Study. The table proves the validity of our three module designs respectively.**

| Model | Sample | Fusion | IAP | CD | F1-score |
|---|---|---|---|---|---|
| A | FPS | - | - | 6.95 | 0.805 |
| A1 | KCPS | - | - | 6.93 | 0.810 |
| B | FPS+KCPS | Spatial | - | 6.72 | 0.819 |
| C | FPS+KCPS | Attn | - | 6.64 | 0.821 |
| D | FPS+KCPS | VA | - | 6.55 | 0.842 |
| Key-Prompt | FPS+KCPS | VA | ✓ | **6.32** | **0.857** |

able to decrease the number of decoder layers without substantial accuracy loss. Consequently, we introduce a lightweight variant of Key-Prompt, equipped with only 5 Transformer decoders compared to AdaPoinTr's 8 decoders, achieving over a 20% reduction in FLOPS and a nearly 1.5 times faster computation, all while incurring a mere 1% dip in accuracy. We compared the model with existing GRNet [34], Seedformer [42], AdaPoinTr [38]. Our lightweight version of Key-Prompt achieves a better result than AdaPoinTr [38] with fewer FLOPs, and our method can process more than 40 point clouds in one second (using a batch size of 1). In our testing environment, Key-Prompt stands out as the fastest among recent models.

**Table 6: Complexity Analysis. Comparisons of GFLOPS and Throughput of our models with other models.**

| Models | FLOPs | Throughput | CD |
|---|---|---|---|
| GRNet [34] | 40.4 G | 32.4 pc/s | 6.74 |
| Seedformer [42] | 53.8 G | 12.7 pc/s | 6.74 |
| AdaPoinTr [38] | 15.1 G | 31.8 pc/s | 6.53 |
| Key-Prompt | 18.2 G | 28.1 pc/s | **6.32** |
| Key-Prompt* | **14.2 G** | **40.2 pc/s** | 6.39 |

Model marked with * represent our light version of Key-Prompt.

## 5 CONCLUSION

In this study, we introduce an innovative architecture named Key-Prompt, focusing on the detailed structures often overlooked by previous methodologies. During the encoding phase, we implement a novel sampling technique known as KCPS. Based on this method, we perform a deep fusion of point cloud features by integrating different sampling methods. Moreover, we designed a unique Semantic-based Interest Area Prompt decoding phase to enable coarse point cloud seed to learn key local details from the input accurately. Our method has achieved state-of-the-art performance on several challenging benchmark datasets through comprehensive experimental validation. Notably, the exceptional generalization ability of our model is underscored in tasks involving few-shot learning and predictions for unseen categories. Additionally, our model exhibits enhanced suitability for real-world applications, highlighting its practical significance and potential for broad adoption.

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

Received 20 February 2007; revised 12 March 2009; accepted 5 June 2009
