# OpenReview forum: "Dig into Detailed Structures: Key Context Encoding and Semantic-based Decoding for Point Cloud Completion"
_acmmm.org/ACMMM/2024/Conference — MM2024 Poster_

### Official Review · Reviewer_R1e1 · 2024-05-23

**Rating:** 4
**Confidence:** 4

**Summary:**

This paper introduces KCPS that preserves richer geometry details via dynamically enforcing larger sampling biases
toward key contour. Integrating KCPS and FPS extractor strategies, the network transform given point clouds into input local
proxies without sacrificing its contour descriptions. The experiments show its effectiveness on several 3D point cloud and 2.5D depth image datasets.

**Strengths:**

+ The performance is significant on several point cloud completion benchmarks.
+ The technical pipeline is clear.

**Limitations:**

My major concern is the novelty of this paper, in which the paper integrates some current methods such KCPS, DGCNN, and e Vector Attention. To better convince me, the authors should explicitly clarify:
1) A summary of why these methods are suitable for your motivation.
2) The challenges when integrating these techniques into your architecture and how you solve these challenges to demonstrate that it's not a simple combination of current methods.

**Suitability:**

2

---

### Official Review · Reviewer_dVoQ · 2024-05-24

**Rating:** 5
**Confidence:** 3

**Summary:**

This paper introduces an innonative point cloud completion method. Key Context Fusion Encoding is proposed to extract and aggregate homologous key context. Semantic-based Decoding is integrated to effectively learn and generate local geometry with semantic correlations from non-nearest neighbors. Experiments on several 3D point cloud and 2.5D depth image datasets have demonstrated the effectiveness of the proposed method.

**Strengths:**

1. This paper proposes a KCPS method which preserves richer geometry details via dynamically enforcing larger sampling biases toward key contour.
2. This paper proposes a semantic-based IAP decoding method that introduces semantics cues into the dependency inference among given and inferred points.
3. Our method achieves state-of-the-art performance on various benchmarks, including 3D point cloud datasets such as PCN, Project shapenet55-34 and KITTI, as well as Redwood RGBD datasets.
4. Good visualization quality also proves the effectiveness of this work.

**Limitations:**

1. Point sampling efficiency is important for most point cloud applications. FPS is known for its high sampling latency. It is better to provide analysis about the complexity of the proposed KCPS.
2. I am curious about the completion on real scene point cloud completion. Only completion results of cars are given, how about sparser categories like pedestrian or cyclist? How about the competion effects on datasets like nuscenes? In real scene, the objects are often very sparse. Hence, can the proposed completion method complete objects with limited points?
3. Figure 2 appears cluttered. Could you please polish it?

**Suitability:**

2

---

### Official Review · Reviewer_v2p9 · 2024-05-26

**Rating:** 4
**Confidence:** 3

**Summary:**

This paper introduces a novel method to enhance 3D point cloud completion. It features Key Context Fusion Encoding, which preserves geometric details by focusing on salient structures and contours, and Semantic-based Decoding, which uses a semantic EdgeConv module to improve local geometry generation. The proposed Key Contour Point Sampling (KCPS) module effectively captures detailed structures. Semantic-based Decoding module uses cross-attention to enhance the generation of detailed and semantically coherent 3D structures from incomplete point clouds. The approach demonstrates superior performance in both qualitative and quantitative evaluations across various datasets.

**Strengths:**

- The Key Context Fusion Encoding method focuses on salient structures and contours, preserving more geometric details compared to traditional methods.
- The Semantic EdgeConv module and Interest Area Prompt (IAP) Decoding improve local geometry generation and ensure semantic coherence, leading to more accurate 3D reconstructions.
- The proposed method outperforms existing state-of-the-art techniques in both qualitative and quantitative evaluations across various datasets.

**Limitations:**

- The method's performance heavily relies on the quality of the input point clouds. Noise or incomplete data in the input would affect the accuracy and quality of the completed point clouds. Please evaluate the robustness of method on the noisy input.

- The Key Contour Point Sampling (KCPS) method, while effective in preserving structural details, may be sensitive to the choice of parameters such as the number of neighbors and the sampling bias, requiring careful tuning for different datasets. Clear guidelines or best practices for tuning these parameters across different datasets would be beneficial.

- Semantic Decoding Challenges: The Semantic-based Interest Area Prompt (IAP) decoding, which involves semantic guidance, may struggle with objects or scenes lacking clear semantic structures, potentially leading to less accurate completions.

**Suitability:**

2

---

### Meta-Review · Area_Chair_4CMj · 2024-06-27

**Recommendation:** Accept (Poster)
**Confidence:** 5

**Metareview:**

This paper propose a novel method for point cloud completion, which shows encouraging performance. All the reviewers agree to accept this manuscript. I recommend a decision of acceptance. In the final version, the authors should include some key results in the rebuttal to address the concerns.